# Evaluation of Optical and Thermal Properties of NIR-Blocking Ophthalmic Lenses Under Controlled Conditions

**DOI:** 10.3390/s25113556

**Published:** 2025-06-05

**Authors:** Jae-Yeon Pyo, Min-Cheul Kim, Seung-Jin Oh, Ki-Choong Mah, Jae-Young Jang

**Affiliations:** 1Department of Optometry, Eulji University, 553 Sanseong-daero, Sujeong-gu, Seongnam-si 13135, Gyonggi-do, Republic of Korea; p961316@naver.com; 2Olens Gangnam Daechi Optical Center, 2936 Nambusunhwan-ro, Gangnam-gu, Seoul 06291, Republic of Korea; 3Department of Optometry, Sungwoon University, 105 Daehak-gil, Sinnyeong-myeon, Yeongcheon-si 38801, Gyeongsangbuk-do, Republic of Korea; metin89@sdc.ac.kr; 4Department of Optometry, Jeonbuk Science College, 117 Chungjeong-ro, Jeongeup-si 56204, Jeonbuk-do, Republic of Korea; osj1314@jbsc.ac.kr; 5International College of Optometry, Jining Polytechnic, 77 Jinyu Road, Jining 272007, China

**Keywords:** near-infrared radiation, near-infrared blocking rate, near-infrared-blocking spectacle lenses, color reproduction, thermal insulation, retinal safety, photothermal effects, optical performance

## Abstract

This study evaluates the optical and thermal performance of near-infrared (NIR)-blocking spectacle lenses at luminous transmittance grades of 0, 2, and 3. Ten lens types were tested, including clear, tinted, and NIR-blocking spectacle lenses (NIBSL). The NIR blocking rate was measured across 780–1100 nm and 1100–1400 nm wavelength bands. Color reproduction was assessed using sharpness (MTF 50), point spread function (PSF), and color accuracy (ΔE_00_) under 1000 lux outdoor illumination. Thermal insulation was analyzed by monitoring porcine skin temperature at 36 °C and 60 °C under each lens type. As a result, the NIBSL showed better near-infrared blocking performance than other types of lenses in both wavelength ranges, and the coated NIBSL blocked near-infrared more effectively than the polymerized lenses. Compared with other types of lenses, NIBSL showed no difference in object identification, color recognition, and reproducibility, so there is no problem in using them together. Strong correlations were observed between lens surface temperature and underlying pig skin temperature, and inverse correlations between NIR blocking rate and pig skin temperature gradient. These findings confirm that NIBSL offer enhanced protection against NIR-induced thermal effects without compromising optical performance, supporting their use in daily environments for ocular and skin safety.

## 1. Introduction

The human eye visually responds to wavelengths within the visible spectrum, enabling the perception of objects. Consequently, it is continuously exposed to various light sources, including sunlight, digital device displays, and artificial indoor lighting. Following the COVID-19 pandemic, daily screen time increased significantly, with global smartphone and laptop usage rising by over 70% and 40%, respectively [1]. Bahkir et al. reported that 94% of users during the pandemic experienced more than 4 additional hours of screen time per day [2]. The American Optometric Association (AOA) notes that prolonged exposure to digital screens (≥2 h/day) can lead to digital eyestrain or computer vision syndrome (CVS) [3].

Solar radiation spans a wide spectral range (290 nm to over 1,000,000 nm), composed of ultraviolet (6.8%), visible light (38.9%), and infrared (54.3%) [4,5]. While UV light poses a high-energy risk, it is mostly absorbed by the outer tissue layers. In contrast, infrared radiation penetrates deeper, with near-infrared (NIR: 780–1400 nm) possessing higher energy than mid- and far-infrared bands [6,7,8]. Multiple studies have reported that NIR exposure increases ocular temperature by 3–5 °C, which can cause inflammation and contribute to cataract development [9,10,11,12]. Furthermore, cumulative exposure may decrease lens transparency and heighten the risk of retinal damage. Since longer NIR wavelengths are more efficiently absorbed by water and lipids, they can penetrate deeply into tissues [13,14], posing additional risks to ocular structures [10,12,15]. To quantify such photothermal risks, the International Commission on Non-Ionizing Radiation Protection (ICNIRP) introduced the Retinal Thermal Hazard Function, R(λ), which spans from the ultraviolet to the near-infrared region [16,17].

To provide a clearer framework for safety evaluation, the International Electrotechnical Commission (IEC) and the Korean Agency for Technology and Standards classify ocular and dermal risks associated with exposure to optical radiation [18,19]. In Korea, NIR-blocking spectacle lenses are commercially available. However, while these products are marketed with protective claims, empirical validation through standardized testing remains insufficient. Although academic interest in NIR-blocking technologies has grown, a lack of standardized evaluation data persists.

In addition to safety considerations, optical performance remains an important criterion when assessing lens quality. One critical factor is color reproduction. Spectacle lenses, whether clear, tinted, or NIR-blocking, may influence color perception. Tinted lenses in particular exhibit varying spectral transmittance depending on dye type and concentration, which can distort perceived color [20,21]. Although tinting can improve visual comfort and reduce glare, side effects such as color distortion must be carefully considered [22]. Although NIR-blocking spectacle lenses are already commercially available, empirical studies evaluating their thermal and optical performance under standardized conditions are scarce. To our knowledge, this is the first study to systematically assess and compare the NIR-blocking efficiency, color accuracy, and thermal shielding performance of lenses across multiple luminous transmittance grades using extended exposure tests. This study seeks to address this gap by offering practical and objective data for both researchers and industry stakeholders.

In this study, we systematically compare the NIR-blocking performance of different spectacle lenses across the 780–1100 nm and 1100–1400 nm bands. Furthermore, color reproduction is assessed under outdoor lighting (1000 lux), and thermal insulation properties are evaluated using porcine skin at 36 °C and 60 °C. All tests were conducted in a temperature- and humidity-controlled chamber to minimize environmental variability. Given the potential risk of photothermal damage from NIR radiation, a comprehensive analysis of blocking efficiency, visual performance, and thermal insulation is essential. To date, most studies evaluating the performance of NIR-blocking spectacle lenses have been limited to single-grade polymerized lenses and short-duration thermal tests, which restrict the generalizability of their findings. This study expands on previous work by systematically evaluating multiple lens types across three ISO luminous transmittance grades (0, 2, and 3) and by conducting extended-duration thermal exposure experiments under controlled conditions. These enhancements aim to establish a more robust empirical basis for the practical use of NIRSL in everyday environments. The findings from this study aim to inform the design and application of NIR-blocking lenses for enhanced visual and physiological safety. All statistical analyses were conducted using IBM SPSS Statistics 27 (SPSS Inc., Chicago, IL, USA)

## 2. Sample Types and Characteristics

Ten different spectacle-lens samples were evaluated in this study (Table 1). These comprised near-infrared-blocker polymerized lenses (NIBPSL) at luminous-transmittance grades 0 and 2; near-infrared-blocker-coated lenses (NIBCSL) at grade 0; near-infrared-blocker-coated + tinted lenses (NIBC + TSL) at grades 0, 2, and 3; clear spectacle lenses (CSL) at grade 0; and tinted spectacle lenses (TSL) at grades 0, 2, and 3. The IR blocking rates for the grade 0 NIBPSL, grade 0 NIBCSL, and grade 0 NIBC + TSL samples were all 30%, while the grade 2 NIBPSL blocked 95% of near-infrared radiation (compared with 30% for grade 2 NIBC + TSL). The grade 3 NIBC + TSL sample also exhibited 30% blocking. Clear (CSL) and tinted (TSL) lenses without any dedicated IR-blocking functionality were included as controls.

All lenses were fabricated with a back-vertex power of 0.00 D (no optical power) and a refractive index of *n* = 1.60. The surfaces of every sample were treated with an anti-reflection coating (UV420). The physical geometry matches commercially available products from SOMO Optical Co. (Seoul, Republic of Korea). To ensure comparable luminous transmittance, the NIBC + TSL and TSL samples were tinted to the same grade levels as the polymerized blocker lenses.

## 3. Experimental Setup and Measurement Methods

To evaluate the influence of ambient temperature on near-infrared blocking ratio and color reproduction, measurements were conducted using the lens samples listed in Table 1 at room temperature (18–25 °C), the Korean summer maximum (36 °C), and an elevated stress-test temperature (60 °C). Thermal insulation performance was assessed separately under the two elevated temperature conditions (36 °C and 60 °C), utilizing thawed porcine skin as a surrogate for human periocular tissue due to its similar thermal conductivity and infrared absorption/emission characteristics. Throughout all experiments, temperature and relative humidity were strictly controlled using a Temperature & Humidity Chamber (TH-PE-100, Jeiotech Co, Daejeon, South Korea).

### 3.1. Near-Infrared Blocking Ratio

Near-infrared light spans wavelengths from 780 nm to 1400 nm, peaking at approximately 1000 nm [23,24]. Although some manufacturers specify blocking rates only up to 1100 nm, others report a single overall percentage. In this study, the spectrum was divided into two regions: 780–1100 nm (including the 1000 nm peak) and 1100–1400 nm (aligned with ISO 13666) [25]. Spectral transmittance, τ(λ), was measured in 10 nm increments using a Cary UV–Visible Spectrophotometer (Cary 5000, Agilent Co, Santa Clara, CA, USA). The solar near-infrared transmittance, τ_IRA_, is calculated as follows:(1)$$\tau_{\mathrm{IRA}}=100\times \frac{1}{1400\;\mathrm{nm}\;-780\;\mathrm{nm}}\int_{780\;\mathrm{nm}}^{1400\;\mathrm{nm}}\tau \left(\lambda \right)•d\lambda \left(\%\right).$$

Here, τ_IRA_ denotes the average transmittance across the 780–1400 nm range (aligned with ISO 13666). The corresponding blocking ratio, B_IRA_, is given by(2)$$B_{\mathrm{IRA}}=100-\tau_{\mathrm{IRA}}\left(\%\right),\;$$
where B_IRA_ represents the percentage of solar near-infrared radiation blocked by the lens.

### 3.2. Color Reproduction

Color reproduction was evaluated outdoors (illuminance ≈ 1000 lux) in accordance with ISO 12233 [26]. Lenses were mounted on a Nikon D-7100 camera (Nikon Co, Tokyo, Japan), and an Imatest SFRplus chart (Imatest Co, Boulder, CO, USA) was imaged to quantify sharpness and the point spread function (PSF). A 24-patch X-Rite ColorChecker chart (Imatest Co., Boulder, CO, USA) was then photographed to assess color accuracy. Captured images were analyzed using the Imatest 4.0 image analysis program (Imatest, Boulder, CO, USA) to extract three key metrics: sharpness (MTF 50 from the SFRplus chart), PSF (major-to-minor axis ratio), and color accuracy (average CIEDE2000 ΔE_00_ across 24 color patches) [27,28,29,30]. Mean values for lenses of luminous-transmittance grades 0, 2, and 3 were compared against clear (CSL) and tinted (TSL) controls. Figure 1 illustrates the standardized imaging setup, incorporating both the SFRplus chart for sharpness and point spread function (PSF) assessment and the 24-patch X-Rite ColorChecker chart for color accuracy evaluation.

### 3.3. Thermal Insulation Performance Evaluation

Thermal insulation performance was assessed under the two elevated temperature conditions (36 °C and 60 °C). Lens samples (near-infrared blocking, clear, and tinted) were mounted 12 mm above thawed porcine skin placed in a Petri dish. Near-infrared illumination was provided by a Philips Infrared-R95E lamp (Philips Co., Katowice, Poland) for three consecutive 60 min exposures (total 180 min). Temperature sensors were placed at the apex of the lens surface and at the geometric center of the pig skin (Figure 2). Real-time temperature data were recorded and segmented into 5 min intervals. Because the baseline skin temperature was not constant across trials, average temperatures at 10, 30, and 60 min were normalized to each trial’s baseline to quantify the thermal insulation performance of each lens. To further investigate the impact of near-infrared blocking performance on thermal insulation, the measured blocking ratios (B_IRA_s) were correlated with corresponding increases in porcine skin temperature.

## 4. Results

### 4.1. Near-Infrared Blocking Ratio

The near-infrared (NIR) blocking performance of the tested lenses was assessed across two spectral bands: 780–1100 nm and 1100–1400 nm.

For the 780–1100 nm range, statistically significant differences were identified among grade 0 lenses (F = 294.497, *p* = 0.001). Coated lenses such as NIBCSL (44.05 ± 11.95%) and NIBC + TSL (43.34 ± 12.31%) exhibited higher blocking efficiency compared to the polymerized NIBPSL (32.35 ± 7.74%) and non-blocking controls, CSL (13.42 ± 3.00%) and TSL (14.22 ± 4.03%) (post hoc: d, e < a < b, c). For grade 2, NIBPSL lenses showed the highest NIR blocking (87.66 ± 21.71%, F = 649.759, *p* = 0.001), followed by NIBC + TSL (44.17 ± 12.47%) and TCL (13.32 ± 3.30%) (post hoc: c < b < a). At grade 3, NIBC + TSL lenses (43.38 ± 12.46%) significantly outperformed TCL lenses (13.12 ± 3.07%, t = 23.450, *p* = 0.001; post hoc: n/a) (Table 2).

In the 1100–1400 nm range, a similar trend was observed. Grade 0 lenses differed significantly (F = 783.684, *p* = 0.001), with NIBCSL (51.31 ± 5.69%) and NIBC + TSL (51.17 ± 5.69%) outperforming NIBPSL (23.15 ± 4.85%), CSL (22.40 ± 4.57%), and TSL (24.62 ± 5.42%) (post hoc: a, d, e < b, c). In grade 2, NIBC + TSL lenses (51.67 ± 5.72%) again showed significantly better performance than NIBPSL (31.08 ± 3.70%) and TCL (23.05 ± 4.61%, F = 899.570, *p* = 0.001; post hoc: c < a < b). At grade 3, NIBC + TSL lenses (51.22 ± 5.69%) significantly outperformed TCL lenses (22.63 ± 4.77%, t = 37.166, *p* = 0.001; post hoc: n/a) (Table 3).

### 4.2. Color Reproduction

#### 4.2.1. Sharpness

Sharpness was quantified using the MTF 50 metric. Statistical analysis showed no significant differences among lenses of the same luminous transmittance grade. For grade 0 lenses, the test statistic was χ^2^ = 8.823 with *p* = 0.082, indicating no statistical significance. Similarly, for grade 2 lenses, χ^2^ = 2.518 and *p* = 0.284, and for grade 3 lenses, Z = −1.360 and *p* = 0.174.

Although not statistically significant, a trend of decreasing sharpness was observed as the NIR blocking grade increased. For example, the average MTF 50 values declined from 2693.94 ± 59.92 in grade 0 NIBPSL to 2466.28 ± 36.12 in grade 2, and further to 2386.54 ± 48.55 in grade 3 NIBC + TSL lenses. This suggests a potential trade-off between increased NIR blocking performance and slight reductions in spatial resolution clarity (Table 4).

#### 4.2.2. Point Spread Function (PSF)

Eccentricity values derived from PSF measurements revealed no statistically significant differences between lens categories within each grade (*p* > 0.05). This suggests that NIR-blocking designs, whether polymerized or coated, do not meaningfully alter point spread characteristics or image distortion compared to control lenses. The average eccentricity values remained close to 0.5 across all lens types and grades (Table 5).

#### 4.2.3. Color Accuracy

The color accuracy of near-infrared-blocking spectacle lenses with a luminous transmittance of grade 0 did not differ statistically significantly from that of clear and tinted spectacle lenses (*p* = 0.486). However, for grade 2 lenses, statistically significant differences were observed (χ^2^ = 21.456, *p* = 0.001). Among them, NIBPSL lenses showed lower ΔE_00_ values, which indicates better color fidelity, compared to both NIBC + TSL and TCL lenses (post hoc: a < b, c). At grade 3, NIBC + TSL lenses also demonstrated better color accuracy than TCL lenses (Z = −2.611, *p* = 0.007; post hoc: a < b) (Table 6).

### 4.3. Thermal Insulation Performance

#### 4.3.1. Pig Skin Temperature

At 36 °C, a statistically significant difference in pig skin temperature was found among grade 0 lenses (F = 6.350, *p* = 0.001), with post hoc analysis indicating that NIBPSL resulted in significantly lower temperature than TSL (a < e). For grade 2 and grade 3 lenses, no statistically significant differences were observed (*p* = 0.838 and *p* = 0.251, respectively) (Table 7).

At 60 °C, a statistically significant difference among grade 0 lenses was observed based on ANOVA (F = 2.900, *p* = 0.023). However, follow-up post hoc comparisons did not identify any significant differences between individual lens pairs (*p*-values > 0.050), indicating that the overall variance did not translate into pairwise statistical significance. For grade 2 lenses, the difference was not significant (*p* = 0.302), while for grade 3 lenses, NIBC + TSL showed significantly lower pig skin temperatures than TCL (t = −2.156, *p* = 0.034) (Table 8).

#### 4.3.2. Gradient of Pig Skin Temperature

Temperature gradients across both 36 °C and 60 °C conditions did not show statistically significant differences between lens types (*p* > 0.05). Nevertheless, lenses with NIR-blocking properties tended to exhibit lower skin temperature gradients relative to clear and tinted lenses. In particular, among grade 0 lenses, coated NIR-blocking lenses (NIBCSL, NIBC + TSL) generally produced smaller gradients than the polymerized NIBPSL, suggesting improved thermal insulation (Table 9 and Table 10).

### 4.4. Correlation Analysis

To examine the thermal relationship between the lens surface and the underlying pig skin, correlation analyses were performed under both 36 °C and 60 °C conditions for all lens grades. As illustrated in Figure 3, strong positive correlations were observed at 36 °C: grade 0 (r = 0.873, *p* = 0.01), grade 2 (r = 0.785, *p* = 0.01), and grade 3 (r = 0.814, *p* = 0.01).

At 60 °C, the correlation remained evident (Figure 4), with grade 0 (r = 0.835, *p* = 0.01), grade 2 (r = 0.679, *p* = 0.01), and grade 3 (r = 0.888, *p* = 0.01), indicating that lens surface temperature is closely associated with heat transfer to the skin.

As shown in Figure 5, negative correlations were found between the near-infrared blocking ratios and pig skin temperature gradients across both the 780–1100 nm (r = −0.564 at 36 °C, *p* = 0.090; r = −0.527 at 60 °C, *p* = 0.117) and 1100–1400 nm ranges (r = −0.358 at 36 °C, *p* = 0.310; r = −0.382 at 60 °C, *p* = 0.276). Although these correlations were not statistically significant (*p* > 0.05), the results suggest a consistent trend: lenses with higher NIR blocking ratios tend to reduce thermal conduction more effectively.

## 5. Discussion

Near-infrared (NIR) rays penetrate the eyes and skin tissues depending on the wavelength, are recognized as heat, and are absorbed by the hemoglobin, myoglobin, bone marrow, and cortex of the human body [31,32,33]. While NIR has therapeutic applications, such as enhancing wound healing, treating tumors, and promoting skin elasticity, it can also pose risks similar to ultraviolet radiation when exposure is prolonged. These risks include anterior eye surface heating, increased tear film evaporation, cataracts, and accelerated skin aging [31,32,33,34,35].

This study assessed NIR blocking performance, color reproduction, and thermal insulation characteristics of ophthalmic lenses across three luminous transmittance grades (0, 2, and 3). Compared to previous studies that used limited exposure durations [36], we conducted extended 60 min trials in a temperature- and humidity-controlled chamber, improving the reliability of thermal insulation evaluation.

In the 780~1100 nm range and the 1100~1400 nm range, polymerized lenses (NIBPSL), coated lenses, and coated + tinted lenses (NIBCSL, NIBC + TSL) showed better performance in blocking near-infrared rays than other types of spectacle lenses (clear and tinted). However, in the wavelength range of 1100 to 1400 nm, grade 0 polymerized lenses (NIBPSL) have lower blocking rates than other types of lenses, so methods to improve the blocking rate should be considered when manufacturing lenses. The blocking performance of NIBSL was significantly higher than other types of lenses, regardless of the manufacturing method and luminous transmittance grade in the 780–1100 nm range, but in the 1100–1400 nm range, the blocking performance of coated lenses and coated + tinted lenses (NIBCSL, NIBC + TSL) was higher than that of polymerized lenses (NIBPSL).

These findings affirm that the lens construction method critically influences NIR blocking performance. For instance, in the 780–1100 nm range, NIBPSL grade 2 lenses showed blocking rates over 87%, while CSL and TSL lenses remained below 15%. Similarly, in the 1100–1400 nm range, NIBCSL and NIBC + TSL lenses maintained blocking rates exceeding 50%, while NIBPSL fell below 25%.

In an outdoor environment (1000 lux), the clarity (MTF 50) and PSF (Eccentricity) of the NIBSL compared to other types of spectacle lenses (clear, tinted) did not differ significantly depending on the luminous transmittance and lens type. The color accuracy (ΔE_00_) did not differ among all spectacle lenses in the 0th grade with high luminous transmittance, and there was no significant difference in grades 2 and 3 with low luminous transmittance, or the NIBSL appeared closer to the original color. The polymerized lens (NIBPSL) was closer to the original color in grade 2 than the coated + tinted lens (NIBC + TSL, TCL), which was consistent with the results of the study by Lee et al., in which the color reproducibility of the grade 2 NIBSL was good [37].

The color accuracy (ΔE_00_) was between 7 and 12, which exceeded the standard (1.5 < ΔE ≤ 3) presented by the display and printing industries that traditionally require precise color reproduction but may still be considered acceptable for general-purpose spectacle lenses, depending on visual sensitivity and application conditions. For tasks requiring precise color discrimination, lenses with lower ΔE_00_, such as grade 2 polymerized or grade 3 coated + tinted lenses, may be preferable. As a result, there is no difference between NIBSL and other types of lenses in terms of color reproducibility evaluation factors such as sharpness, point spread function, and color accuracy, so there will be no problem using them together.

Thermal insulation results showed that NIR-blocking lenses reduced pig skin temperature more effectively than clear or tinted lenses. At 36 °C, NIBPSL grade 0 lenses reduced pig skin temperature by an average of 2.60 °C compared to TSL lenses. At 60 °C, NIBC + TSL grade 3 lenses showed a 1.56 °C lower temperature compared to TSL. These trends were consistent across grades 0, 2, and 3, with coated lenses generally performing better than polymerized variants.

A strong correlation was found between the lens surface and pig skin temperatures at 36 °C and 60 °C, and an inverse correlation was found between the NIR blocking rate and the pig skin temperature gradient, supporting the conclusion that high NIR blocking reduces heat transfer. For example, the correlation between lens surface temperature and pig skin temperature reached r = 0.873 for grade 0 lenses at 36 °C. Likewise, negative correlations were observed between the NIR blocking rate in the 780–1100 nm and 1100–1400 nm ranges and the corresponding pig skin temperature gradients.

While this study primarily focused on the empirical evaluation of NIR-blocking spectacle lenses under practical conditions, it does not include theoretical modeling such as thin-film interference analysis, which may further elucidate the underlying optical mechanisms. Due to proprietary constraints regarding lens coating composition and design, access to such detailed specifications was limited. Future studies involving collaboration with manufacturers could help bridge this gap and integrate mechanism-based interpretation into performance assessments. These findings validate the functional benefits of NIBSL for mitigating infrared-induced thermal stress. Their utility extends beyond vision protection to potential applications in ocular and dermal thermal safety, particularly in high-radiation environments. Further public health and clinical attention to NIR exposure, alongside broader adoption of NIBSL, may help reduce risks of chronic thermal damage to sensitive ocular structures and periocular skin.

## 6. Conclusions

This study evaluated the near-infrared (NIR) blocking performance, color reproduction, and thermal insulation properties of ophthalmic lenses with different luminous transmittance grades. The experimental results demonstrated that NIR-blocking spectacle lenses, particularly those with coated and coated + tinted constructions, showed significantly higher blocking efficiency compared to clear and tinted control lenses in both the 780–1100 nm and 1100–1400 nm wavelength ranges. Polymerized lenses exhibited relatively lower blocking performance in the longer-wavelength range (1100–1400 nm), highlighting the importance of lens fabrication methods.

The optical characteristics, including sharpness (MTF 50), point spread function (PSF), and color accuracy (ΔE_00_), showed no statistically significant differences among lens types of the same grade, confirming that NIR-blocking lenses do not impair visual performance in practical environments. While all tested lenses exhibited ΔE_00_ values exceeding 3, these values remain within ranges that may still be considered acceptable, depending on the application and observer sensitivity.

In thermal insulation assessments, NIR-blocking spectacle lenses reduced the temperature increase of pig skin more effectively than control lenses, with coated lenses generally providing better thermal protection than polymerized ones. Strong positive correlations were observed between lens surface temperature and pig skin temperature, while inverse correlations were observed between NIR blocking ratios and thermal gradients.

These findings confirm that NIR-blocking spectacle lenses can be used in daily life without compromising optical clarity or color perception. Moreover, they contribute to thermal safety by attenuating infrared radiation that may lead to ocular or dermal stress. Further studies involving human-related models or clinical trials may be necessary to validate long-term protective effects. The study provides essential data supporting the broader application of NIR-blocking spectacle lenses for enhanced comfort and protection in environments with significant infrared exposure.

## Figures and Tables

**Figure 1 sensors-25-03556-f001:**
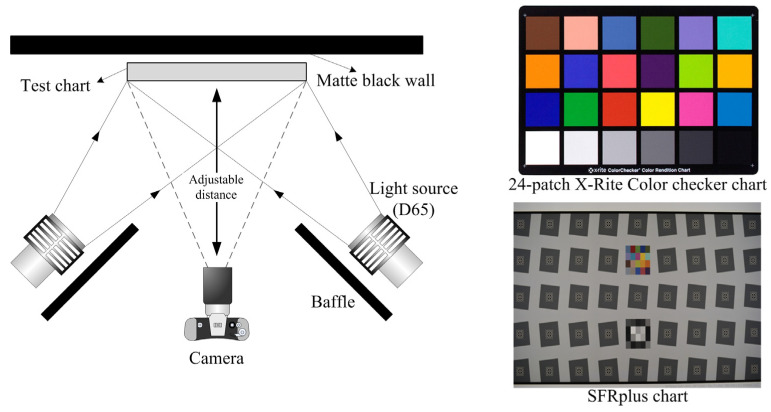
Experimental setup for color accuracy and image quality assessment. The test system consists of a matte black wall, a test chart, two D65 light sources, and baffles to prevent direct light into the camera. The SFRplus chart was used to evaluate sharpness and PSF, while the 24-patch X-Rite ColorChecker chart was used to assess color accuracy.

**Figure 2 sensors-25-03556-f002:**
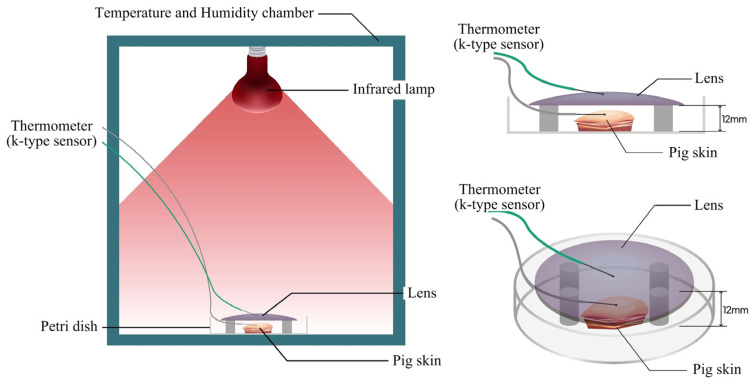
Experimental setup for thermal insulation evaluation using porcine skin. The setup includes a temperature and humidity chamber, an infrared (IR) lamp, and k-type thermocouples for real-time temperature monitoring. The lens was positioned 12 mm above the pig skin, with both placed inside a Petri dish to ensure consistent measurement geometry. Diagrams show cross-sectional and top-perspective schematic views of the experimental configuration.

**Figure 3 sensors-25-03556-f003:**
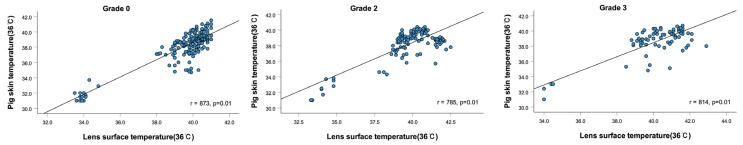
Correlation between lens surface temperature and pig skin temperature (36 °C).

**Figure 4 sensors-25-03556-f004:**
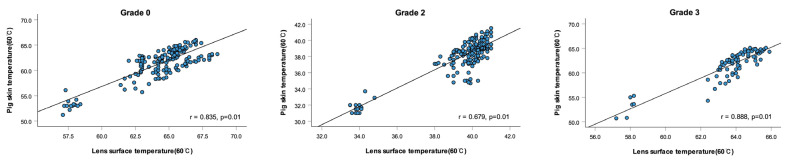
Correlation between lens surface temperature and pig skin temperature (60 °C).

**Figure 5 sensors-25-03556-f005:**
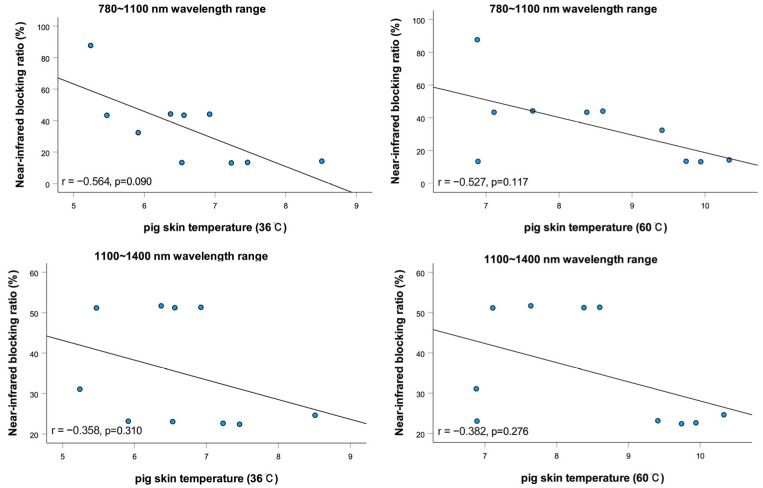
Correlation between near-infrared blocking rate and pig skin temperature (36 °C, 60 °C).

**Table 1 sensors-25-03556-t001:** Lens sample types, luminous-transmittance grades, and near-IR blocking rates.

Sample No.	Lens Type	Grade (Luminous Transmittance)	IR Blocking Rate (%)
1	NIBPSL ^1^	0	30
2	NIBPSL ^1^	2	95
3	NIBCSL ^2^	0	30
4	NIBC + TSL ^3^	0	30
5	NIBC + TSL ^3^	2	30
6	NIBC + TSL ^3^	3	30
7	CSL ^4^	0	- *
8	TSL ^5^	0	- *
9	TSL ^5^	2	- *
10	TSL ^5^	3	- *

* “-“ indicates no dedicated IR-blocking functionality. ^1^ Near-infrared-blocker polymerized spectacle lens. ^2^ Near-infrared-blocker-coated spectacle lens. ^3^ Near-infrared-blocker-coated + tinted spectacle lens. ^4^ Clear spectacle lens. ^5^ Tinted spectacle lens.

**Table 2 sensors-25-03556-t002:** NIR blocking ratio (780~1100 nm).

Lens Category	Grade	NIR Blocking Ratio (%)	Test Statistic	*p*-Value	Post Hoc (Scheffe)
NIBPSL (a)	0	32.35 ± 7.74	F = 294.497	0.001	d, e < a < b, c
NIBCSL (b)	0	44.05 ± 11.95			
NIBC + TSL (c)	0	43.34 ± 12.31			
CSL (d)	0	13.42 ± 3.00			
TSL (e)	0	14.22 ± 4.03			
NIBPSL (a)	2	87.66 ± 21.71	F = 649.759	0.001	c < b < a
NIBC + TSL (b)	2	44.17 ± 12.47			
TCL (c)	2	13.32 ± 3.30			
NIBC + TSL	3	43.38 ± 12.46	t = 23.450	0.001	n/a
TCL	3	13.12 ± 3.07			

**Table 3 sensors-25-03556-t003:** NIR blocking ratio (1100~1400 nm).

Lens Category	Grade	NIR Blocking Ratio (%)	Test Statistic	*p*-Value	Post Hoc (Scheffe)
NIBPSL (a)	0	23.15 ± 4.85	F = 783.684	0.001	a, d, e < b, c
NIBCSL (b)	0	51.31 ± 5.69			
NIBC + TSL (c)	0	51.17 ± 5.69			
CSL (d)	0	22.40 ± 4.57			
TSL (e)	0	24.62 ± 5.42			
NIBPSL (a)	2	31.08 ± 3.70	F = 899.570	0.001	c < a < b
NIBC + TSL (b)	2	51.67 ± 5.72			
TCL (c)	2	23.05 ± 4.61			
NIBC + TSL	3	51.22 ± 5.69	t = 37.166	0.001	n/a
TCL	3	22.63 ± 4.77			

**Table 4 sensors-25-03556-t004:** Sharpness of tested lenses.

Lens Category	Grade	Sharpness (MTF 50, lw/ph)	Test Statistic	*p*-Value
NIBPSL	0	2693.94 ± 59.92	χ^2^ = 8.823	0.082
NIBCSL	0	2638.74 ± 58.59		
NIBC + TSL	0	2635.56 ± 81.36		
CSL	0	2692.62 ± 41.62		
TSL	0	2673.08 ± 66.48		
NIBPSL	2	2466.28 ± 36.12	χ^2^ = 2.518	0.284
NIBC + TSL	2	2431.94 ± 71.49		
TCL	2	2403.47 ± 131.27		
NIBC + TSL	3	2386.54 ± 48.55	Z = −1.360	0.174
TCL	3	2369.02 ± 50.74		

**Table 5 sensors-25-03556-t005:** Point spread function (PSF) of tested lenses.

Lens Category	Grade	Major Axis	Minor Axis	Eccentricity	Test Statistic	*p*-Value
NIBPSL	0	4.33 ± 0.90	3.39 ± 0.31	0.50 ± 0.25	χ^2^ = 1.786	0.775
NIBCSL	0	4.17 ± 0.59	3.40 ± 0.20	0.50 ± 0.21		
NIBC + TSL	0	4.19 ± 0.46	3.37 ± 0.27	0.54 ± 0.17		
CSL	0	4.00 ± 0.34	3.34 ± 0.21	0.50 ± 0.18		
TSL	0	4.33 ± 0.82	3.51 ± 0.33	0.52 ± 0.27		
NIBPSL	2	4.32 ± 0.69	3.56 ± 0.37	0.49 ± 0.21	χ^2^ = 1.909	0.385
NIBC + TSL	2	3.96 ± 0.36	3.46 ± 0.40	0.44 ± 0.17		
TCL	2	4.03 ± 0.38	3.48 ± 0.45	0.44 ± 0.19		
NIBC + TSL	3	3.95 ± 0.34	3.43 ± 0.34	0.44 ± 0.17	Z = −0.882	0.378
TCL	3	4.03 ± 0.42	3.57 ± 0.40	0.41 ± 0.18		

**Table 6 sensors-25-03556-t006:** Color accuracy of tested lenses.

Lens Category	Grade	Color Accuracy (ΔE_00_)	Test Statistic	*p*-Value	Post Hoc (Bonferroni)
NIBPSL	0	8.69 ± 2.22	χ^2^ = 3.446	0.486	n/a
NIBCSL	0	7.25 ± 0.97			
NIBC + TSL	0	7.48 ± 0.63			
CSL	0	7.11 ± 1.52			
TSL	0	7.37 ± 0.59			
NIBPSL (a)	2	9.52 ± 0.87	χ^2^ = 21.456	0.001	a < b, c
NIBC + TSL (b)	2	11.55 ± 0.49			
TCL (c)	2	11.60 ± 0.61			
NIBC + TSL	3	10.51 ± 0.41	Z = −2.611	0.007	n/a
TCL	3	10.97 ± 0.33			

**Table 7 sensors-25-03556-t007:** Pig skin temperature at 36 °C.

Lens Category	Grade	Pig Skin Temp (36 °C)	Test Statistic	*p*-Value	Post Hoc (Bonferroni)
NIBPSL (a)	0	36.96 ± 2.02	F = 6.350	0.001	a < e
NIBCSL (b)	0	37.96 ± 1.99			
NIBC + TSL (c)	0	37.83 ± 1.70			
CSL (d)	0	38.37 ± 2.43			
TSL (e)	0	39.31 ± 2.40			
NIBPSL	2	38.09 ± 1.81	F = 0.177	0.838	n/a
NIBC + TSL	2	38.17 ± 2.26			
TCL	2	38.36 ± 2.11			
NIBC + TSL	3	38.20 ± 2.13	t = −1.156	0.251	n/a
TCL	3	38.78 ± 2.31			

**Table 8 sensors-25-03556-t008:** Pig skin temperature at 60 °C.

Lens Category	Grade	Pig Skin Temp (60 °C)	Test Statistic	*p*-Value	Post Hoc (Bonferroni)
NIBPSL	0	61.08 ± 3.15	F = 2.900	0.023	n/a
NIBCSL	0	60.96 ± 2.60			
NIBC + TSL	0	61.57 ± 3.01			
CSL	0	60.92 ± 3.01			
TSL	0	62.87 ± 3.20			
NIBPSL	2	60.78 ± 2.33	F = 1.209	0.302	n/a
NIBC + TSL	2	61.75 ± 2.94			
TCL	2	61.56 ± 3.37			
NIBC + TSL	3	60.85 ± 3.24	t = −2.156	0.034	n/a
TCL	3	62.43 ± 3.22			

**Table 9 sensors-25-03556-t009:** Gradient of pig skin temperature at 36 °C.

Lens Category	Grade	Gradient (36 °C)	Test Statistic	*p*-Value
NIBPSL	0	5.91 ± 1.43	χ^2^ = 8.023	0.091
NIBCSL	0	6.92 ± 0.73		
NIBC + TSL	0	5.47 ± 0.84		
CSL	0	7.46 ± 1.42		
TSL	0	8.51 ± 1.09		
NIBPSL	2	5.24 ± 0.39	χ^2^ = 3.467	0.177
NIBC + TSL	2	6.37 ± 1.15		
TCL	2	6.53 ± 0.85		
NIBC + TSL	3	6.56 ± 1.06	Z = −0.655	0.513
TCL	3	7.23 ± 1.18		

**Table 10 sensors-25-03556-t010:** Gradient of pig skin temperature at 60 °C.

Lens Category	Grade	Gradient (60 °C)	Test Statistic	*p*-Value
NIBPSL	0	9.41 ± 2.00	χ^2^ = 3.100	0.541
NIBCSL	0	8.60 ± 1.39		
NIBC + TSL	0	7.11 ± 2.34		
CSL	0	9.74 ± 2.90		
TSL	0	10.33 ± 2.02		
NIBPSL	2	6.88 ± 1.34	χ^2^ = 1.067	0.587
NIBC + TSL	2	7.64 ± 1.24		
TCL	2	6.89 ± 1.07		
NIBC + TSL	3	8.38 ± 1.60	Z = −1.528	0.127
TCL	3	9.94 ± 1.33		

## Data Availability

The original contributions presented in this study are included in the article. Further inquiries can be directed to the corresponding authors.

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
