# Peer review of "Evaluation of Optical and Thermal Properties of NIR-Blocking Ophthalmic Lenses Under Controlled Conditions"

_sensors, 2025, doi:10.3390/s25113556_

Round 1

Reviewer 1 Report

Comments and Suggestions for Authors

This study compares the optical and thermal performance of various spectacle lenses, focusing on near-infrared (NIR) blocking lenses with different luminous transmittance grades. Ten types of lenses, including clear, tinted, and NIR blocking spectacle lenses (NIBSL), were tested for their ability to block NIR radiation across two wavelength ranges. The lenses were also evaluated for color accuracy and sharpness under bright outdoor conditions. Thermal insulation was assessed by measuring temperature changes on porcine skin beneath each lens type. Results show that NIBSL, especially coated versions, block NIR more effectively without affecting visual performance or color recognition. The lenses also reduce skin heating caused by NIR, suggesting they provide better protection for eyes and skin in everyday settings.

  1. How does the NIR blocking performance differ between coated and polymerized NIBSL lenses?

  2. Were there any noticeable differences in user comfort or visual experience when wearing the different lens types outdoors?

  3. How might these findings translate to human skin and eye protection in real-world environments?

  4. Could the study’s temperature monitoring on porcine skin accurately predict long-term thermal effects in humans?

  5. Are there any potential trade-offs in lens durability or cost associated with the improved NIR blocking coatings?

  6. How do these NIBSL lenses perform under varying lighting conditions, such as indoor or low-light environments?

  7. What are the implications for people who require high precision in color recognition, such as artists or professionals working with color-critical tasks?

  8. Can this lens technology be integrated into other types of eyewear or protective gear beyond spectacles?

Reviewer 2 Report

Comments and Suggestions for Authors

This manuscript systematically evaluated the optical performance (transmittance, MTF 50, PSF, chromatic aberration) and thermal protection effect (temperature gradient of pigskin) of various near-infrared blocking spectacle lenses (NIBSL). The data is detailed, the structure is complete, and the conclusion is clear. The research confirmed the advantages of coated NIBSL in near-infrared blocking and heat insulation performance, and it did not have a negative impact on visual recognition function. This finding has clear reference value for engineering applications in the eyewear industry, such as the design of sunglasses lenses. 

However, from the perspective of academic research, this paper has the following limitations: 

The scientific questions and lack of innovation: The research is essentially a performance comparison test of multiple groups of lenses, without proposing new scientific hypotheses or mechanism-based analyses (for example: how coating/materials regulate the quantitative relationship between near-infrared blocking rate and thermal effect), nor addressing controversial issues within the field. 

The methodological depth is lacking: the data presentation lacks statistical validation (such as error analysis and significance testing), and is not associated with existing theories or models (such as the influence of optical film interference on near-infrared reflection). The experimental design does not go beyond the scope of conventional industrial testing. 

Limited incremental contribution: Although the results confirmed the practicality of NIBSL, they did not clearly demonstrate its breakthrough compared to existing research (for example, a comparison of its performance with similar lenses in the literature), or propose new design guidelines. 

In conclusion, this paper is more of a technical test report than a research paper. Although it has significant engineering significance, its academic contribution does not meet the journal's requirements for "original research". It is suggested that the authors supplement the following content to enhance its scientific nature: 

Pose clear scientific questions (such as "Can a certain type of coating structure balance light transmittance and heat insulation performance?"); 

Enhance the analysis of the physical/chemical mechanisms behind the data; 

By comparing with existing literature, the distinctive value of this research is highlighted.

Round 2

Reviewer 1 Report

Comments and Suggestions for Authors

i am happy with revisions ...

Reviewer 2 Report

Comments and Suggestions for Authors

Although the manuscript did not discuss the related scientific issues due to patent avoidance, it was the first to systematically evaluate and compare the near-infrared blocking efficiency, color accuracy, and thermal shielding performance of lenses with different light transmittance grades. The manuscript fills the relevant gap and provides practical and objective data for researchers and industry stakeholders. Given its potential data originality, the paper can be considered for acceptance.